# Alteration of Skin Sympathetic Nerve Activity after Pulmonary Vein Isolation in Patients with Paroxysmal Atrial Fibrillation

**DOI:** 10.3390/jpm12081286

**Published:** 2022-08-05

**Authors:** Wei-Ting Sung, Li-Wei Lo, Yenn-Jiang Lin, Shih-Lin Chang, Yu-Feng Hu, Fa-Po Chung, Jo-Nan Liao, Ta-Chuan Tuan, Tze-Fan Chao, Chin-Yu Lin, Ting-Yung Chang, Ling Kuo, Chih-Min Liu, Shin-Huei Liu, Wen-Han Cheng, An Khanh-Nu Ton, Chu-Yu Hsu, Chheng Chhay, Ahmed Moustafa Elimam, Ming-Jen Kuo, Pei-Heng Kao, Wei-Tso Chen, Shih-Ann Chen

**Affiliations:** 1Heart Rhythm Center, Department of Medicine, Division of Cardiology, Taipei Veterans General Hospital, Taipei 11220, Taiwan; 2Institute of Clinical Medicine and Cardiovascular Research Center, National Yang Ming Chiao Tung University, Taipei 11221, Taiwan; 3Cardiovascular Center, Taichung Veterans General Hospital, Taichung 40705, Taiwan; 4National Chung Hsing University, Taichung 40227, Taiwan

**Keywords:** atrial fibrillation, skin sympathetic nerve activity, pulmonary vein isolation, autonomic system

## Abstract

Autonomic system plays a pivotal role in the pathogenesis of paroxysmal atrial fibrillation (AF). Skin sympathetic nerve activity (SKNA) is a noninvasive tool for assessing sympathetic tone. However, data on changes in SKNA after ablation are limited. Here, we retrospectively enrolled 37 patients with symptomatic drug-refractory paroxysmal AF who underwent pulmonary vein isolation (PVI) with radiofrequency ablation (RFA) or cryoablation (CBA). SKNA was measured from the chest and right arm 1 day prior to ablation, as well as 1 day and 3 months after ablation. One day after ablation, the SKNA-Arm increased from 517.1 µV (first and third quartiles, 396.0 and 728.0, respectively) to 1226.2 µV (first and third quartiles, 555.2 and 2281.0), with an increase of 179.8% (125% and 376.0%) (*p* < 0.001); the SKNA-Chest increased from 538.2 µV (first and third quartiles, 432.9 and 663.9) to 640.0 µV (first and third quartiles, 474.2 and 925.6), with an increase of 108.3% (95.6% and 167.9%) (*p* = 0.004), respectively. In those without recurrence, there was a significant increase in SKNA 1 day after ablation as compared with those before ablation. Twelve patients received SKNA measurement 3 months after ablation; both SKNA-Arm (*p* = 0.31) and SKNA-Chest (*p* = 0.27) were similar to those before ablation, respectively. Among patients with symptomatic drug-refractory paroxysmal AF receiving PVI, increased SKNA was observed 1 day after ablation and returned to the baseline 3 months after ablation. Elevation of SKNA was associated with lower early and late recurrences following ablation.

## 1. Introduction

Atrial fibrillation (AF) is the most common cardiac arrhythmia, which may lead to thromboembolic events and increased mortality rate. The pulmonary vein (PVs) has been well known as a common source of focal ectopy for AF. Pulmonary vein isolation (PVI) using radiofrequency ablation (RFA) is a class IA recommendation for the treatment of drug-refractory symptomatic paroxysmal AF and a class IIA recommendation for some patients with persistent AF [1,2].

In addition to the PV, the autonomic nervous system also plays a substantial role in AF initiation and maintenance. Paravertebral cervical and thoracic ganglia are responsible for sympathetic innervation of the heart. Among them, the stellate ganglion is the most crucial one [3]. In a canine model, Choi et al. showed that intrinsic and extrinsic cardiac nerve activities were closely related to each other, and the initiation of AF was associated with activation of the intrinsic cardiac autonomic nervous activity [4].

Direct measurement of satellite ganglion nerve activity in a canine model is feasible [5], but it is not clinically practical in humans due to the invasiveness. Skin has abundant sympathetic innervation and the sympathetic nerve cell bodies in the skin of the upper limbs are also located at the middle cervical, thoracic, and stellate ganglia, which is similar to those responsible for cardiac sympathetic innervations [5,6]. Because nerve structures are extensively connected, it is possible that sympathetic nerve activity may activate simultaneously as an increase in cardiac sympathetic activity. It has been proven that subcutaneous skin sympathetic nerve activity and superficial skin sympathetic nerve activity (SKNA) closely correlated with stellate ganglion activity in a canine model [7]. Moreover, SKNA has been proven to be correlated with changes in the sympathetic tone of humans [8,9].

Regarding AF, an observational study has shown that the average SKNA measurements were higher at the onset and termination as compared with that in sinus rhythm. It was also suggested that there was a negative correlation between AF episodes and average SKNA, probably due to dysfunction of the sinoatrial node [10]. However, data on changes in SKNA after AF ablation are limited. In this study, we aimed to investigate changes in SKNA and its relationship to the outcome after percutaneous catheter ablation therapy in paroxysmal AF.

## 2. Materials and Methods

### 2.1. Patient Population

A total of 37 patients with symptomatic drug-refractory paroxysmal AF who underwent radiofrequency catheter ablation (RFA) or cryoablation (CBA) were retrospectively enrolled. The definition of paroxysmal AF was defined according to the statement from the 2017 Heart Rhythm Society Expert Consensus [11]. Echocardiography was performed within 1 month prior to the ablation. Informed consent forms were given to the patients. The study was approved by the Institutional Review Board of Taipei Veteran General Hospital (IRB number 2021-08-013BC).

### 2.2. Catheter Ablation

#### 2.2.1. Electrophysiological Study and Mapping Strategy

The details of the ablation method have been described in our previous works [12,13,14]. In brief, all the patients discontinued antiarrhythmic medications for more than 5 half-lives (except for amiodarone) before the procedure. Each patient underwent electrophysiological study and ablation while in the fasting non-sedated state. Local anesthesia was used in all patients.

#### 2.2.2. RFA

In the patients presenting to the lab with sinus rhythm, we first tried to identify spontaneous AF using burst atrial pacing or isoproterenol up to 4 µg/min. After a successful trans-septal procedure, PV isolation was performed guided by the EnSite Velocity system using an irrigated ablation catheter (TactiCath™ or FlexAbility™, St. Jude Medical, Little Canada, MN, USA), by the Carto 3 system using an irrigated ablation catheter (Thermocool™, Biosense Webster, Diamond Bar, CA, USA), or by the Rhythmia system using an irrigated ablation catheter (Intellanav™, Boston Scientific, Marlborough, MA, USA). The choice for circumferential or segmental PVI depended on physicians’ decisions. If residual PV potentials were found, supplementary ablation applications were applied along the circumferential lines close to the earliest ipsilateral PV potentials.

Successful PVI was demonstrated by obtaining the entrance and exit blocks of the PVs, the absence of any electrical activity inside PVs, or dissociated PV activity during sinus rhythm. If AF did not stop after PVI, we identified non-PV triggers. In our laboratory, we had a standard strategy for identification of non-PV triggers which had been published in our previous literatures [2,13,15,16,17].

In brief, the location of the non-PV trigger was identified by evaluation of the activation sequence of the high right atrium (RA), His bundle area, and coronary sinus (CS). A duodecapolar catheter (1 mm electrode length and 2 mm interelectrode spacing) was placed into the superior vena cava (SVC) and atriocaval junction area to detect SVC triggers. The time interval between the high RA and the His bundle area during sinus rhythm and ectopy differentiated the site of ectopy as the SVC, upper crista terminalis, or PVs. Simultaneous mapping of the SVC and the right PVs was done to clarify the true initiating foci. If the earliest activation site was in the interatrial septum (IAS), simultaneous mapping of the right and left septa was performed. To map non-PV triggers from the left atrium, the activation time interval between the proximal and distal pairs of the CS catheter during ectopy was evaluated. If the earliest activation site was in the vicinity of the left PV ostium or posterolateral portion of the mitral annulus, ligament of Marshall potentials were clarified by differential pacing or epicardial mapping. For patients with non-PV triggers, catheter ablation toward the earliest electrical activity or a local unipolar QS pattern of the ectopic beat preceding the onset of AF was performed. The end point of non-PV trigger ablation was disconnection between the SVC and the RA, the CS and the RA, or elimination of other non-PV ectopic beats with negative provocation of AF.

The decision of performing linear ablation was based on the operator. Ablation other than PVI was defined as additional ablation in this study. After AF ablation, cavotricuspid isthmus (CTI) ablation was routinely performed. Bidirectional conduction block of the CTI was confirmed after restoration to sinus rhythm.

#### 2.2.3. Cryoablation (CBA)

The details of the CBA have been described in our previous literature [15,18]. In short, a 28 mm second-generation cryoballoon (Arctic Front Advance, Medtronic Inc., Minneapolis, MN, USA) was advanced to the left atrium, inflated, and kept in the PV ostium for each vein. Contrast injection was given, and total contrast retention with no backflow to the atrium confirmed optimal vessel occlusion. During cryoablation over the right PV, diaphragmatic stimulation was performed with a quadripolar catheter positioned in the superior vena cava and pacing the ipsilateral phrenic nerve with a 1000 ms cycle and 20 mA output to avoid phrenic nerve palsy. If AF was still noted or could be induced after cryoablation, electrical cardioversion was performed to restore sinus rhythm. Non-PV triggers were searched and ablated using the 4 mm non-irrigated ablation catheter (Boston Scientific, Marlborough, MA, USA) using the mapping method mentioned above in the RFA section. CTI ablation was routinely performed with the 8 mm non-irrigated ablation catheter (Boston Scientific, Marlborough, MA, USA) at the end of the CBA procedure.

### 2.3. Signal Recording of Skin Nerve Activity

After admission to the ward, SKNA was documented via the MP36 system (BIOPAC Systems Inc., Goleta, CA, USA) one day prior to the ablation. All signals were recorded with standard ECG electrodes. The configuration is shown and described in Figure 1. SKNA-Chest was recorded from the Lead I configuration of the ECG leads. The signals were obtained with a sampling rate of 2000 Hz for at least 10 min. The signal recorded from the Lead I configuration was low-pass filtered between 0.05 Hz and 150 Hz to display ECG, and high-pass filtered between 500 Hz to 1000 Hz to display SKNA-Chest. The SKNA recorded from the right arm with the same technique was documented as SKNA-Arm.

One day after ablation, both SKNA-Chest and SKNA-Arm were repeated for at least 10 min. Analysis of the signals was performed using the BIOPAC Student Lab Software. A total of 600 s were selected in each patient to calculate mean SKNA-Chest and mean SKNA-Arm. Three months after ablation, SKNA measurement was repeated using the same method at the clinic. Twelve out of the 37 patients returned to the clinic and agreed to follow-up SKNA measurement.

### 2.4. Follow-Up

Follow-up at the outpatient clinic was arranged within 1 month after the ablation, and then every 1–3 months. Resting surface 12-lead electrocardiogram, 24-hour Holter monitoring, and/or 1 week duration of cardiac event recording were used to assess if there was a recurrence. The definition of early and late recurrences after the ablation was based on the 2017 Heart Rhythm Society Expert Consensus [11]. An early recurrence was defined as any recurrence of AF > 30 s during the first 3 months of follow-ups; a late recurrence was defined as any AF episode >30 s noted between 3 and 12 months.

### 2.5. Statistics

Continuous variables were expressed as mean (first and third quartiles) or mean ± standard deviation, and categorical variables as counts (percentages). For differences between the two groups, numerical data were analyzed using the Mann–Whitney U test and categorical data using the two-sided Fisher’s exact test. The percentage of SKNA change was calculated via dividing the SKNA after ablation by the SKNA before ablation. The percentage of SKNA change was calculated for each patient and the mean was obtained based on them. For the comparisons among the SKNA measurements before, after, and 3 months after the ablation, the Wilcoxon test was used. The level of statistical significance was set at a *p*-value <0.05. All the analyses were performed using commercial IBM SPSS software.

## 3. Results

### 3.1. Baseline Characteristics and Ablation Results

The baseline characteristics of all patients are shown in Table 1. Five (13.5%) of 37 patients showed AF upon admission, and 18 (48.6%) patients were under anti-arrhythmic drugs during the blanking period at discharge.

The ablation results are shown in Table 2. Ablation was successful among all patients. None of the patients in the CBA group received additional ablation, while five patients in the RFA group received additional ablation, including superior vena cava isolation, mitral isthmus ablation, left atrial appendage ridge ablation, coronary sinus ostium ablation, left atrial roof ablation, and focal ablation of the coronary sinus. Early recurrence occurred in three (13%) patients in the RFA group, and one (7.1%) patient in the CBA group; late recurrence was seen in five (21.7%) patients in the RFA group and two (14.3%) patients in the CBA group. There was no significant difference in early and late recurrence rate between the two groups.

### 3.2. SKNA Results

#### 3.2.1. SKNA One Day before Ablation Procedure

The SKNA-Arm and SKNA-Chest of all the patients before ablation are shown in Table 3. The results were presented as median (first and third quartiles). At baseline, there were no differences in both SKNA measurements between the CBA and RFA groups. The baseline SKNA-Arm was 543.8 µV (first quartile, 397.0 and third quartile 745.24) in the RFA group and 479.1 µV (first quartiel, 394.0 and third quartile 671.4) in the CBA group (*p* = 0.53); the baseline SKNA-Chest was 556.8 µV (first quartile 457.0 and third quartile 734.0) in the RFA group and 734.0 µV (first quartile 420.7 and third quartile 605.2) in the CBA group (*p* = 0.14).

#### 3.2.2. SKNA One Day after the Ablation

##### All Patients

The changes in SKNA-Arm and SKNA-Chest one day after ablation are listed in Table 3. Thirty-one (83.8%) and 26 patients (70.3%) showed increases of SKNA-Arm and SKNA-Chest after ablation, respectively. As compared to those before ablation, there were significant increases in both SKNA-Arm and SKNA-Chest one day after the ablation procedure, respectively. The mean HR before ablation was 70.32 ± 15.17 bpm and 70.86 ± 10.85 bpm after ablation (*p* = 0.296). There was no significant difference between the resting HR.

As comparing with those before ablation in the RFA group, there was a 150.3% (first quartile, 98.8% and third quartile 346.8%) increase in the SKNA-Arm (*p* = 0.002), and a 109.3% (first quartile, 94.7% and third quartile, 172.5%) increase in the SKNA-Chest (*p* = 0.014). As for the CBA group, the mean SKNA-Arm increased significantly but not the SKNA-Chest. As compared with those before ablation, there was a 187.5% (first quartile 147.0% and third quartile 443.9%) increase in the SKNA-Arm (*p* = 0.001), and a 106.3% (first quartile, 96.4% and third quartile, 126.7%) increase in the SKNA-Chest (*p* = 0.084).

##### Patients with/without Early Recurrence

The results are listed in Table 3. The baseline SKNA did not differ between those with and without early recurrence (*p* = 0.33 for SKNA-Arm and *p* = 0.35 for SKNA-Chest). For those without early recurrence, both SKNA-Arm and SKNA-Chest increased significantly 1 day after ablation as compared with those 1 day prior to ablation (*p* < 0.001 for SKNA-Arm and *p* = 0.006 for SKNA-Chest, respectively). No significant elevation in both SKNA measurements could be seen in the early recurrence group (*p* = 0.465 in SKNA-Arm and *p* = 0.465 in SKNA-Chest).

##### Patients with/without Late Recurrence

The results are listed in Table 3. The baseline SKNA did not differ between those with and without late recurrence (*p* = 0.69 for SKNA-Arm and *p* = 0.14 for SKNA-Chest). For those without late recurrence, both SKNA-Arm and SKNA-Chest increased significantly 1 day after ablation as compared with those 1 day prior to ablation (*p* < 0.001 for SKNA-Arm and *p* = 0.003 for SKNA-Chest). No significant elevation in both SKNA measurements could be seen in the late recurrence group (*p* = 0.063 in SKNA-Arm and *p* = 0.499 in SKNA-Chest).

#### 3.2.3. SKNA 3 Months after Ablation

The results are shown in Table 4. The SKNA measurements 3 months after ablation were similar to the baselines in both SKNA-Arm (*p* = 0.31) and SKNA-Chest (*p* = 0.27). The change in SKNA is depicted in Figure 2.

## 4. Discussion

### 4.1. Main Findings

In our study, we found that: First, the mean SKNA increased after ablation in patients with paroxysmal AF who received either RFA or CBA. We speculated this result was associated with the elimination of cardiac parasympathetic tone caused by the ablation procedure, leading to the subsequent elevation of systemic sympathetic tone. Second, the significant increases in SKNA measurements were associated with lower early and late recurrences, suggesting neuromodulation of the parasympathetic system by ablation was a possible explanation for the association. Third, SKNA returned to the baseline 3 months after the ablation procedure, suggesting that the neuromodulation effects of AF ablation were temporary and would not cause long-term cardiac autonomic imbalance.

### 4.2. Neuromodulation Effect of PVI

Several studies have suggested that ablation could attenuate cardiac parasympathetic activity [19,20], while data directly measuring cardiac sympathetic tone have been lacking. Instead, heart rate variability (HRV) parameters such as standard deviate of the normal–normal intervals (SDNN) and proportion of normal–normal intervals differing from their neighbors by >50 milliseconds (pNN50) were used to represent both sympathetic and parasympathetic activities. Decreased SDNN and pNN50 have both been reported after AF ablation [20,21]. However, the parameter for sympathovagal balance (i.e., low frequency/high frequency ratio (LF/HF)) remained unchanged 1 day after ablation, which implicated the balance of vagal and sympathetic tone were not altered by the procedure [21,22]. These findings suggested both cardiac vagal and sympathetic tones might be changed simultaneously after ablation.

Previous studies in the literature have found that systemic sympathetic activity was reciprocally related to cardiac parasympathetic activity [23,24]. On the one hand, in healthy subjects, muscle sympathetic activity (MSNA) increased during tilting, but the HRV parameter indicating parasympathetic activity (i.e., high frequency, HF) decreased [23]. On the other hand, in patients with heart failure who received exercise therapy, the HRV parameter HF increased after training while MSNA decreased [24]. Using immunostaining, Tan et al. found that there was no specific area of adrenergic or cholinergic predominance around pulmonary veins, with over 25% of nerve fibers containing both components [25]. These findings indicated that ablation targeting only sympathetic or parasympathetic nerves was almost impossible. Cui et al. found that MSNA rose 1 day after ablation [21]. In our study, we also found neuromodulation effects following AF ablation. As compared with the baseline, SKNA increased significantly 1 day after ablation. We hypothesized that ablation could cause local damage to the post-ganglionic cardiac parasympathetic neurons and sympathetic nerve ending, while the sympathetic neurons remained intact at the stellate ganglion. A decrease in cardiac parasympathetic tone might lead to elevation of SKNA, which stands for systemic activation of the sympathetic tone.

Second, we found that SKNA returned to its baseline 3 months after ablation, and this was compatible with previous studies [19,26,27,28]. Using HRV as a surrogate for autonomic activity, Pappone et al. and Pokushalov et al. both found that a change in HRV could be observed immediately after ablation, and returned to the baseline at 6 months after ablation [19,26]. In addition to HRV, S100B protein, a biomarker for neuronal growth released immediately after neural damage, has been found to be elevated along with high sensitivity troponin I immediately after AF ablation [27]. In a study of ganglion plexus (GP) ablation in a canine model, Wang et al. found the nerve density in GPs was significantly downregulated 1 month after GP ablation, but returned to the baseline after 6 months of ablation. The marker protein of nerve sprouting, GAP43, was not upregulated until 6 months after GP ablation, indicating the time needed for axon regeneration [28]. In our study, we found the phenomenon of nerve activity retuning to the baseline occurred as early as 3 months after ablation. This may be associated with the neural regeneration effects. When cardiac parasympathetic tone arises as nerves grow, the systemic sympathetic activity (i.e., SKNA) reciprocally decreases.

### 4.3. Neuromodulation Effect and Ablation Outcome

Pappone et al. found that vagal reflexes elicited during PVI caused an increase in HRV that implicated vagal withdrawal, and those with recurrences of AF demonstrated a limited increase in HRV after ablation. Among those without recurrence, the increase in HRV lasted for at least 3 months before attenuation [19]. Significantly reduced parasympathetic activity after ablation was found to be associated with a lower recurrence rate [22]. Scherschel et al. demonstrated that a significant elevation of S100B protein immediately after PVI was associated with a lower recurrence rate of AF within 6 months after ablation, indicating a stronger neuromodulation effect [27]. In our study, instead of looking into the parasympathetic system, we used a noninvasive method to evaluate the sympathetic activity during the peri-operative period. We found that an elevation of SKNA one day after ablation correlated with a lower recurrence rate, for both early and late recurrences. We hypothesized that the significantly elevated SKNA was a reflection of adequate attenuation of cardiac parasympathetic activity caused by the ablation procedure. Some subjects in our study did not have increased SKNA after ablation, a possible explanation may be due to the inter-individual variation of autonomic nerve distribution [29]. However, the small sample size in this study could cause the lack of statistical significance of SKNA measurements in those with early or late recurrence.

## 5. Conclusions

Using a noninvasive modality, in this study, we demonstrated the neuromodulation effects of AF ablation from the viewpoint of systemic sympathetic tone. Significant increases in both SKNA-Arm and SKNA-Chest were observed 1 day after ablation and were associated with lower early and late recurrences. After 3 months of follow-up, both SKNA measurements returned to their baselines. The results provided insight into the alternation of the autonomic system brought by catheter ablation.

## Figures and Tables

**Figure 1 jpm-12-01286-f001:**
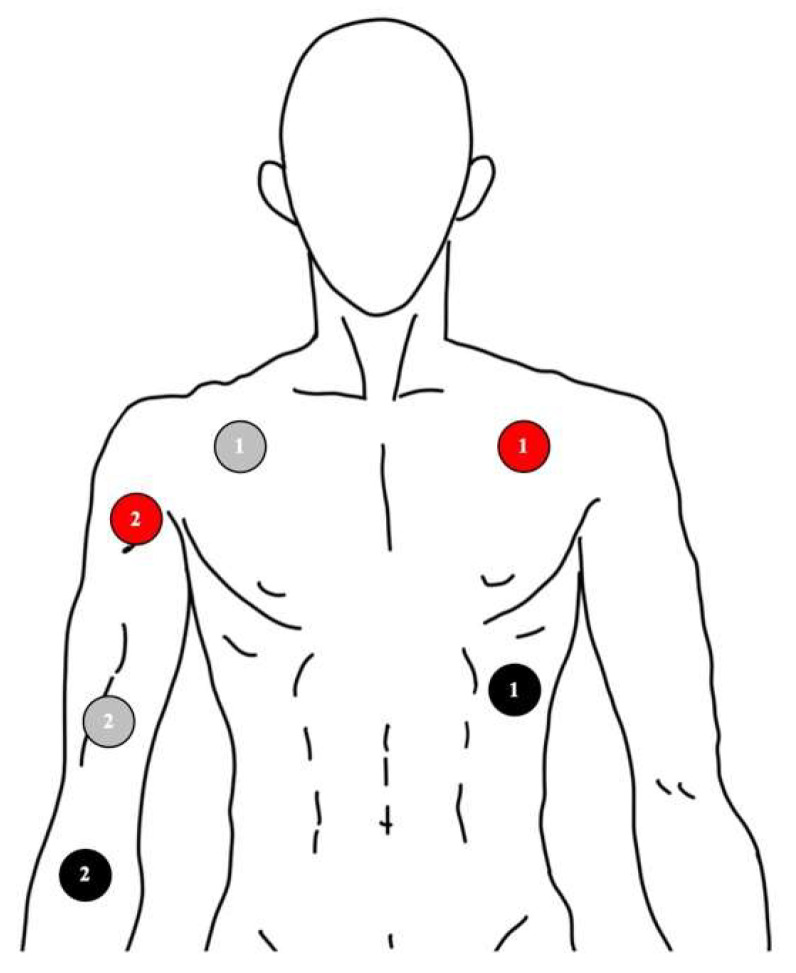
Schematic illustration of the configuration of ECG leads. The numbers of the electrodes correspond to each channel. The grey color stands for the negative electrode, the red color stands for the positive electrode, and the black color stands for the reference electrode. Channel 1 records the ECG (electrocardiography) from the negative electrode in the right subclavian area to the positive electrode in the left, and the reference electrode is placed in the left abdomen area. The signal is transformed to SKNA (skin nerve activity)-Chest via computer software. Channel 2 records the SKNA from the right arm to avoid ECG contamination.

**Figure 2 jpm-12-01286-f002:**
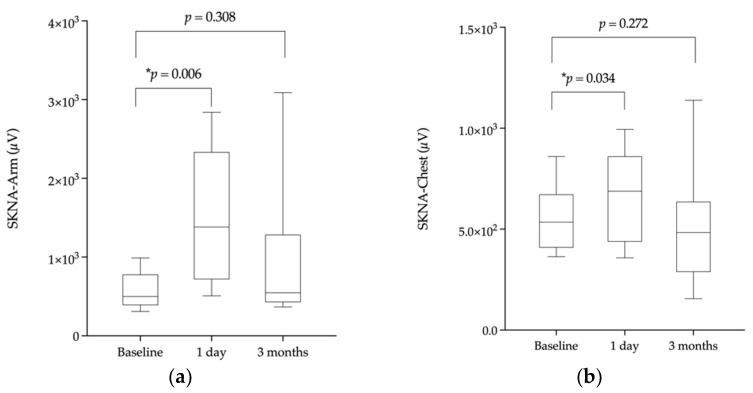
The box plot depicting skin nerve activity (SKNA) at baseline, 1 day after ablation, and 3 months after ablation. A box contains from the first quartile to the third quartile, and the segment inside shows the median. The whiskers above and below the box show the 90% percentile and 10% of percentile of the distribution, respectively. The asterisk represents *p* that is less than 0.05. (**a**) The change of SKNA-Arm over times; (**b**) the change of SKNA-Chest over times.

**Table 1 jpm-12-01286-t001:** Baseline characteristics in all patients (*n* = 37).

	Results
Male, *n* (%)	34 (91.9%)
Age (years old), mean ± SD	58.9 ± 9.0
LVEF (%), mean ± SD	57.8 ± 6.1
LAD (mm), mean ± SD	39.4 ± 4.8
RA enlargement, *n* (%)	5 (14.3%)
Hypertension, *n* (%)	15 (40.5%)
Diabetes mellitus, *n* (%)	3 (8.1%)
CAD, *n* (%)	5 (13.5%)
HFrEF, *n* (%)	3 (8.1%)

CAD, coronary artery disease; HFrEF, heart failure with reduced ejection fraction; LAD, left atrium dimension; LVEF, left ventricular ejection fraction; RA, right atrium; SD, standard deviation.

**Table 2 jpm-12-01286-t002:** Ablation results and outcome.

Ablation Type	
RFA, *n* (%)	23 (62.2%)
CBA, *n* (%)	14 (37.8%)
Follow up time (days), mean ± SD	973.0 ± 229.7
Circumferential PVI, *n* (%)	28 (75.7%)
Segmental PVI, *n* (%)	9 (24.3%)
RSPV, *n* (%)	37 (100%)
RIPV, *n* (%)	35 (94.6%)
LSPV, *n* (%)	37 (100%)
LIPV, *n* (%)	35 (94.6%)
Complete isolation of PV, *n* (%)	37 (100%)
Additional ablation, *n* (%)	5 (13.5%)
Early recurrence, *n* (%)	4 (10.8%)
Late recurrence, *n* (%)	7 (18.9%)

CBA, cryo-balloon ablation; LIPV, left inferior pulmonary vein; LSPV, left superior pulmonary vein; PV, pulmonary vein; PVI, pulmonary vein isolation; RFA, radiofrequency ablation; RIPV, right inferior pulmonary vein; RSPV, right superior pulmonary vein; SD, standard deviation.

**Table 3 jpm-12-01286-t003:** Comparisons of mean SKNA between before and after the ablation (*n* = 37).

	Before	After	*p*-Value	Percentage of Change
Median SKNA-Arm (Q1 and Q3) (µV)				
All	517.1 (396.0; 728.0)	1226.2 (555.2; 2281.0)	<0.001	179.8% (125.0%, 376.0%)
With early recurrence	682.6 (470.4; 923.9)	892.7 (539.0; 1936.3)	0.465	136.5% (91.1%;318.5%)
Without early recurrence	481.5 (394.7; 708.1)	1338.9 (555.2; 2281.0)	<0.001	204.2% (125.0%; 376.0%)
With late recurrence	517.1 (396.8; 705.3)	531.0 (462.4; 1510.7)	0.063	147.0% (92.2%; 346.8%)
Without late recurrence	503.3 (395.0; 777.1)	1438.0 (671.4; 2480.3)	<0.001	187.5% (125.3%; 446.1%)
Mean SKNA-Chest (Q1; Q3) (µV)				
All	538.2 (432.9; 663.9)	640.0 (474.2; 925.6)	0.004	108.3% (95.6%; 167.9%)
With early recurrence	474.3 (408.1; 569.7)	497.4 (469.7; 697.7)	0.465	118.1% (97.3%; 145.9%)
Without early recurrence	544.1 (437.7; 707.0)	658.9 (474.2; 970.6)	0.006	108.1% (95.6%; 167.9%)
With late recurrence	625.9 (544.1; 819.2)	596.4 (509.4; 1040.9)	0.499	116.1% (77.6%; 124.7%)
Without late recurrence	520.6 (420.5; 643.6)	649.5 (448.8; 905.8)	0.003	108.2% (98.4%; 172.5%)

ECG, electrocardiography; Q1, first quartile; Q3, third quartile; SKNA, skin nerve activity.

**Table 4 jpm-12-01286-t004:** Comparisons of mean SKNA among before, 1 day, and 3 months after the ablation (*n* = 12).

	Median SKNA-Arm (Q1; Q3) (µV)	*p*-Value *	Median SKNA-Chest (Q1; Q3) (µV)	*p*-Value *
Before ablation	446.7 (396.9; 680.3)		535.1 (420.7; 663.9)	
1 Day after ablation	1660.4 (978.9; 2439.7)	0.006	688.2 (443.7; 854.2)	0.034
3 Months after ablation	546.4 (428.4; 1109.6)	0.308	483.7 (306.0; 613.7)	0.272

* *p* value obtained by comparing the SKNA post ablation to SKNA before ablation. Q1, first quartile; Q3, third quartile; SKNA, skin nerve activity.

## Data Availability

The datasets analyzed in this study are available from the corresponding author upon reasonable request.

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
