# Peer review of "Alteration of Skin Sympathetic Nerve Activity after Pulmonary Vein Isolation in Patients with Paroxysmal Atrial Fibrillation"

_jpm, 2022, doi:10.3390/jpm12081286_

Round 1
Reviewer 1 Report
This topic is very close to me, and I have read this article with great pleasure.
But I have a few concerns, which I would like to list.
I have some big methodological questions:
1. The authors point out that the study was retrospective. Therefore, it is very important to describe in detail why one group of patients underwent RFA and the other CBA. Were there any differences between these groups?
2. I also wonder if this is a retrospective study, does this mean that the study of SKNA was conducted by the authors on a routine basis? And so, the authors took this data retrospectively? Is the study of SKNA performed in all patients with AF before RFA/CBA in your clinic?
3. If 12 people were examined after 3 months, why did the authors include 37 patients in the article? After all, it turns out that only 12 people had long-term outcomes and SKNA values?
4. As for statistical analysis, I think the use of mean ± standard deviation when describing the values of SKN A is not applicable. There is clearly an abnormal distribution of data, since SD is almost equal in value to mean, and sometimes even greater than it. For example (line 190): 234.8% ± 435.9%. This is mathematically incorrect. It is necessary to use the Interquartile range (Q1;Q3) and use only nonparametric methods.
5. One of the important points in the article is described in the conclusion (line 217-219):
"Second, the significant increases of the SKNAs were associated with lower early and late recurrences, suggesting the neuromodulation of parasympathetic system by ablation was the possible explanation for the association."
However, such data were not presented in the article. And even on the contrary, the authors received a lack of connection of SKNA with early recurrence/late Recurrence. I didn't quite understand this point.
Minor revision:
6. In the Table 1. Male, age and LVEF are repeated twice in this table. The values of the "Comorbidities" column are empty.
7. Line 169. The mean SKNA-Armand SKN A-Chest of all the patients before ablation are shown in the Table 2. There is no such data in the table 2.
8. Line 181. The changes of mean SKNA-Arm and SKNA-Chest 1 day after ablation are listed in Table 2. Again, there is no such data in the table 2. Authors need to be more attentive.
9. Table 4. The level of statistical significance should be reflected in the table for clarity. These data are in the text.
The method of studying the vegetative balance of the skin nerves (which mainly play a role in the thermoregulation of the skin as a protective barrier from the external environment) is very questionable and has not yet received general recognition. Moreover, the accepted methods of analyzing the vegetative balance (such as heart rate variability), in many studies, were in no way related to the parameters of SKNA.
Despite this, fundamental research in this area is very interesting.
However, methodologically correct studies are needed to obtain convincing data.
Reviewer 2 Report
The authors describe an interesting study using non-invasive assessment of skin sympathetic nerve activity to descriptively follow changes in sympathetic tone post-pulmonary vein ablation. The findings support current understanding of the pathophysiology of atrial fibrillation, and as noted in the discussion have been identified by other authors, albeit with other measures of autonomic activity. As a result, the findings lack some novelty, but remain interesting.
The manuscript is generally well-written, but retains some grammatical errors and structural errors (see specific comments below) that will require closer proof-reading.
General comments:
1. The authors describe noticeable changes in SKNA in all patients and those without early/late recurrence. It would be interesting to note whether similar changes in SKNA occur in patients with recurrence, or whether changes are blunted.
2. A large number of patients did not provide 3 month data, which unfortunately limits the utility of this time point. Why did this occur?
3. It would also be interesting to see trends of SKNA with and without recurrence at the 3 month time point as well. A graph depicting trends over time (baseline, post ablation, 3 months) might also be a nice way to present the data.
4. It would be interesting to see if changes in SKNA can predict either early or late recurrence. If a difference exists between SKNA levels with/without recurrence, ROC analyses could be considered.
5. Are there other markers of sympathetic changes that could be presented, e.g. resting HR post-ablation?
Specific comments:
1. Table 1 - there is repetition of data in the last 3 rows that should be removed.
2. Section 3.2.1 - the SKNA results are not in Table 2, but in Table 3. This should be rectified.
3. Table 3 - I am not certain how the percentage of change values are obtained. E.g. For all mean SKNA-Arm change, the difference (After - Before)/Before = 161%.
Reviewer 3 Report
This study investigates the difference of the skin sympathetic nerve activity (SKNA) between pre- and post-ablation conditions in patients with atrial fibrillation (AF). The authors found an increase of SKNA obtained on the chest and the right arm one day after ablation.
Although this is a very nice study and the findings are interesting, there are some problems in the presentation, wording and phrasing. Here are some of the examples:
In Abstract, "AF" and "PVI" should be spelled out even when done so in Introduction section. "Compare to before ablation" seems strange.
In Introduction, "Paravertebral cervical and thoracic ganglia 'were' responsible" looks odd.
In Materials and Methods, "If non-PV triggers were found, non-PV triggers ablation would be performed" but the authors should state this more clearly. The words "SKNA-Chest" and "SKNA-Arm" should be explained in the section even though they are understandable on Fig. 1.
In Results, three lines at the bottom are the same as those at the top on Table 1. "The mean SKNA-Arm and SKNA-Chest of all the patients before ablation are shown in" Table 3, not Table 2.
In Discussion, I don't understand why "mean SKNAs 'would' increase", which suggests that the authors are not sure if they really increased or not.
Finally I feel very strange that >20 co-authors did not pointed out these issues.
Round 2
Reviewer 1 Report
The authors have corrected all the comments quite qualitative.
Author Response
Thank you very much for your encouraging suggestions and comments.
Reviewer 2 Report
General comments:
The authors have addressed the comments provided in the initial review satisfactorily.
Some grammatical corrections are still required throughout the manuscript. The abstract, for example, includes redundant phrase repetitions that could be removed.
In the abstract, the interquartile range for the mean SKNA-Arm after ablation is defined as "first interquartile; firth interquartile" (line 26). There is probably a typo with "firth" = "third", as defined later in section 2.5.
In the methods section 2.2.2 the contraction "NPV" is used without defining it, and is not used consistently after. It should be expanded to "non-PV" or properly defined and used consistently throughout the paragraph.
Regarding statistical analysis, if variables are treated as non-normal (i.e. SKNA), would recommend describing measure of central tendency as median as opposed to mean (which should be reserved for normal data sets). This would make more descriptive sense to use median +/- IQR vs. mean +/- SD.
Section 3.2.2.2, 3.2.2.3 - the authors have included data regarding SKNA changes in those with early/late recurrence, as suggested. Reviewer would suggest including the numerical results in Table 3 instead of in-text. This would facilitate comparison of all results and p values and reduce amount of text needed.
The authors conclude that an increase in SKNA 1 day post-ablation is associated with fewer recurrences. The data suggests a lack of significant increase in SKNA in those with recurrence, although the numbers included are small and the lack of difference might be due to a lack of statistical power. Some recognition of this limitation of small sample size should be included in the discussion.
Author Response
Thank you very much for the encouraging comments. The responses to those comments are below and those changes have been made into the revised manuscript.
- Some grammatical corrections are still required throughout the manuscript. The abstract, for example, includes redundant phrase repetitions that could be removed.
Response: Thank you for the comment. We have deleted some redundant words in the revised manuscript (page 1, line 20-35)
- In the abstract, the interquartile range for the mean SKNA-Arm after ablation is defined as "first interquartile; firth interquartile" (line 26). There is probably a typo with "firth" = "third", as defined later in section 2.5.
Response: Thank you very much and we are sorry for our mistakes. We have made the corrections in the revised manuscript (page 1, line 26).
- In the methods section 2.2.2 the contraction "NPV" is used without defining it, and is not used consistently after. It should be expanded to "non-PV" or properly defined and used consistently throughout the paragraph.
Response: Thank you for the comment and we are sorry for our mistake. The abbreviation “NPV” was replaced by “non-PV” in the revised manuscript (page 3, line 121 and line 127).
- Regarding statistical analysis, if variables are treated as non-normal (i.e. SKNA), would recommend describing measure of central tendency as median as opposed to mean (which should be reserved for normal data sets). This would make more descriptive sense to use median +/- IQR vs. mean +/- SD.
Response: Thank you very much for the comment. The descriptions of all SKNAs in the revised manuscript have been changed to median ± interquartile range.
- Section 3.2.2.2, 3.2.2.3 - the authors have included data regarding SKNA changes in those with early/late recurrence, as suggested. Reviewer would suggest including the numerical results in Table 3 instead of in-text. This would facilitate comparison of all results and p values and reduce amount of text needed.
Response: Thank you for the comment. We have added the numerical results in Table 3. The changes were made in the revised manuscript (page 6, Table 3).
- The authors conclude that an increase in SKNA 1 day post-ablation is associated with fewer recurrences. The data suggests a lack of significant increase in SKNA in those with recurrence, although the numbers included are small and the lack of difference might be due to a lack of statistical power. Some recognition of this limitation of small sample size should be included in the discussion.
Response: Thank you for the comment. We have added the limitation in the revised manuscript (page 9, line 443-444)